Improving synthetic media generation and detection using generative adversarial networks

Zia Rabbia 1
Rehman Mariam 1 mariamrehman@gcuf.edu.pk
Hussain Afzaal 1
Nazeer Shahbaz 1
Anjum Maria 2
1 Department of Information Technology , Government College University Faisalabad, Punjab , Pakistan
2 Department of Computer Science, Lahore College for Women University , Lahore, Punjab , Pakistan
Alatas Bilal
Electronic publication date: 2024 Sep 20
Publication date: 2024
Volume: 10
Electronic Location ID: e2181
Received 2024 Feb 2; Accepted 2024 Jun 17
Copyright: © 2024 Zia et al.
Copyright year: 2024
Copyright holder: Zia et al.
License: This is an open access article distributed under the terms of the Creative Commons Attribution License, which permits unrestricted use, distribution, reproduction and adaptation in any medium and for any purpose provided that it is properly attributed. For attribution, the original author(s), title, publication source (PeerJ Computer Science) and either DOI or URL of the article must be cited.
License URL: https://creativecommons.org/licenses/by/4.0/

Keywords: Generative adversarial networks, Deep neural networks, Image manipulation, DeepFake, Manipulation detection

Funding: The authors received no funding for this work.

==============================
Synthetic images ar­­­e created using computer graphics modeling and artificial intelligence techniques, referred to as deepfakes. They modify human features by using generative models and deep learning algorithms, posing risks violations of social media regulations and spread false information. To address these concerns, the study proposed an improved generative adversarial network (GAN) model which improves accuracy while differentiating between real and fake images focusing on data augmentation and label smoothing strategies for GAN training. The study utilizes a dataset containing human faces and employs DCGAN (deep convolutional generative adversarial network) as the base model. In comparison with the traditional GANs, the proposed GAN outperform in terms of frequently used metrics i.e., Fréchet Inception Distance (FID) and accuracy. The model effectiveness is demonstrated through evaluation on the Flickr-Faces Nvidia dataset and Fakefaces d­­ataset, achieving an FID score of 55.67, an accuracy of 98.82%, and an F1-score of 0.99 in detection. This study optimizes the model parameters to achieve optimal parameter settings. This study fine-tune the model parameters to reach optimal settings, thereby reducing risks in synthetic image generation. The article introduces an effective framework for both image manipulation and detection.

Introduction

In recent years, there has been a growing concern regarding the increasing threat posed by manipulated images, resulting in alterations to individuals’ identities (Rehman, Rasool & Safder, 2023). The alterations in face characteristics, or other visual aspects can lead to alterations in the content (Alanazi, Ushaw & Morgan, 2024). They might vary from little adjustments to major transformations, influencing the perceived identity and authenticity of the persons represented in the image. Generative adversarial networks (GANs), powered by artificial intelligence, are a major source of content generation for Synthetic media (Raza, Munir & Almutairi, 2022; Siegel et al., 2024).

Synthetic media has been linked to cybercrimes like identity theft, fraud, and the spread of false information, despite its potential for entertainment (Sudhakar & Shanthi, 2023). Deepfake detection is a challenging task that requires advancement in digital forensics (Rana et al., 2022). The traditional face tampering is time-consuming and needs certain software and knowledge. Image synthesis has become increasingly accessible and efficient due to advancements in deep learning. Deepfake videos and other types of altered media have proliferated on numerous social media sites and internet platforms as a result of shift. The combination of technology and malicious intent highlights the importance of strong countermeasures against the dangers of misrepresented media are needed (Jung, Kim & Kim, 2020). The primary goal of this research is to tackle challenges associated with reliability of images to improve the robustness of the detection model.

Generative adversarial networks training remains a challenging task due to the problem of mode collapse which occurs when the generator focuses on generating a single mode rather than collecting all potential modes in the dataset (Qian et al., 2020; Allahyani et al., 2023). The recent advances in CNN-based generative models which significantly enhanced the visual quality of generated images (Salvi et al., 2023). Generative adversarial networks type of deep generative models have demonstrated success in manipulated content generation and detection. GAN’s basic structure consists of two components i.e., a Generator and a Discriminator.

The Generator attempts to fool the Discriminator which distinguishing between real and fake images (Li et al., 2021c). The adaptability of GANs, incorporating data from various fields, has resulted in the development of multiple GAN variations i.e., Vanilla GAN (Goodfellow et al., 2020), Deep Convolutional GAN (Radford, Metz & Chintala, 2016), Conditional GAN (Isola et al., 2017) and semi-Supervised GAN (Salimans et al., 2016).

The application of GANs has resulted in significant progress on a variety of image synthesis challenges. These networks have shown the ability to generate synthetic images with improved resolution by implementing adversarial loss. They are designed to generate a variety of realistic graphics that closely follow real data distributions using conventional distribution technique (Simion, Radu & Florea, 2024). The facial attribute editing in these models involves decoding the latent representation of a given facial image based on the desired attributes (Ke & Wang, 2023). Differentiating between digitally generated and manipulated images, the DAD-HCNN technique tackles the misuse of image-generation technology, especially GANs (Jain et al., 2020).

Data augmentation is essential in the training of deep neural networks particularly ones with many parameters (Shorten & Khoshgoftaar, 2019). In the tasks of image classification, object identification, semantic segmentation and several augmentation approaches have been presented and widely used. To choose the appropriate augmentation technique for GANs in the context of image manipulation can be a challenging task (Wang, She & Ward, 2021). The advanced GANS models frequently limit augmentation to core operations such as random crop and flip to avoid the addition of unwanted noise components. The research modifies the GAN architecture to enhance its adaptability for both synthetic media generation and detection tasks. Simultaneously reducing the occurrence of mode collapse and improving the model’s generalization capabilities to distinguish between real and fake images. The proposed model outperforms in terms of generating more realistic images when compared to standard models, and identifying fake human appearances. This study employs a manual search approach to identify optimal values for model parameters, with the goal of producing the most effective output.

In summary, the study contributions are as follows: The study contributes in term of improving DCGAN architecture through the integration of data augmentation techniques and the implementation of label smoothing strategies. The addition of a 10% dropout layer within the discriminator block further improved the model’s performance.

The proposed model has been designed to improve the discrimination between real and fake images. The model aims to reduce the occurrence of mode collapse problem in the generated output by achieving lower FID score and accuracy compared to the other GAN.

To optimize the model’s performance by performing an extensive parameters tuning to achieve optimal results. This entails adjusting various parameters such as the optimizer, batch size, latent vector, and learning rate to maximize the effectiveness of the model.

The structure of this article is outlined as follows: “Related work” section contains thorough review of existing studies. “Materials and Methods” section contains detailed description of the proposed framework. “Results” section contains detailed results and model evaluation. “Conclusion and Future work” contains conclusion, study limitation and future work to further improve this work.

Related work

The facial manipulation techniques significantly improved image-generating technologies. They easily modify information by replacing one person’s identity or facial expression with another, leading to the rise of “deepfakes.” Accurately identifying fake images is essential to preventing their use and manipulation. This study examines how well the Vision Transformer (ViT) detects deep fake images. They uses global feature extraction approaches along with patch-wise self-attention modules to improve accuracy. The proposed technique obtains a 99.5% to 100% detection accuracy, precision, recall, and F1 rate. When compared to other models, the ViT architecture performs better. Future work will assess the model’s performance in real-world contexts. Additionally, the deepfakes problem will be investigated as a multiclass classification (Arshed et al., 2023).

GAN with message importance measure (MIM-GANs) effectively generate more diverse samples. They combine numerous generators, a classifier, and a discriminator. Classifier is responsible for generating statistical probabilities that indicate the possibility of generated data falling into particular categories. These probability distributions demonstrate which data clusters are efficiently created by the separate generators. They use mutual information to calculate the correlation between generators. The unique approach uses mutual information as a constraint that can help generators in learning separate data clusters. They perform a systematic evaluation of the mutual information values between each generator. These estimated values are then applied to the generator loss function, resulting in a changed loss function. They use back-propagation to update the settings of each generator to minimize mutual information while also minimizing generator loss. The tests were performed to evaluate MIM-GAN’s efficacy utilizing benchmark datasets like MNIST, CIFAR10, and CelebA. MIM-GAN achieved the better FID Score, CelebA 186.528, CIFAR10 203.954, MNIST 7.958, and generated varied data at various resolutions demonstrating considerable performance gain (Li et al., 2021b).

This article proposed a method for detecting face swapping and other identity manipulations in single images. Face swapping methods, such as deepFake, modify the face region while maintaining the context’s appearance unchanged. The study demonstrates that this method creates inconsistencies between the two regions, which serve as detectable indicators of manipulation a context recognition network that examines the facial surroundings, such as the hair, ears, and neck, and a face identification network that concentrates on the face region defined by accurate semantic segmentation. The method achieves state-of-the-art results, with 99.7% accuracy on the FF-DF benchmark and 66.0% accuracy on the Celeb-DF-v2 benchmark for face manipulation detection (Nirkin et al., 2022).

They introduced STARGAN which is capable of assisting image-to-image translations across many domains utilizing a unified framework. This capability includes the ability to change features such as color and facial expressions (Karras, Laine & Aila, 2021). They presented an innovative approach that aims to tackle the challenges associated with the training instability to improve image generation in GANs. DM2GAN is a hybrid generative model that integrates auto-encoders to learn a low-dimensional latent representation of data and performs distribution matching in both data and latent space. The auto-encoder captures important data features and distribution patterns which could be fine-tuned to learn a structured latent representation of the data. GANs extracts high-dimensional data space. The author evaluated their model on various datasets, including Toy 2D, CIFAR-10, and MNIST-1K, demonstrating its ability to simplify complex data and uncover meaningful patterns. DM2GAN achieved a remarkable FID score 20.7 (Zuo et al., 2022). They proposed model to train GAN and to improve GAN generation capabilities to solve the mode collapse problem. The study proposed IDGAN a novel technique that combines a unique loss function and an image classification network as an auxiliary discriminator to simplify adversarial training among three main components i.e., Generator a Discriminator and an auxiliary Discriminator.

They improve the stability of IDGAN training by employing an incremental learning method that gradually refines the generator. D2 fine-tunes image classification network such as ResNet and DenseNet which improves at discriminating between real and fake images. When compared to previous GANs, IDGAN outperforms in terms of Inception score and FID score 32.8 on CIFAR-10 and 48.1 on STL-10 datasets. Additionally, the experiments on the CelebA-HQ dataset shows that IDGAN achieve remarkable image-generating capabilities, even at high resolutions (Baek, Yoo & Bae, 2019).

The study solved the difficulties in developing an appropriate mapping function across different domains. GANs have shown greater capabilities in solving this challenge in both supervised and unsupervised environments. However, a fundamental practical difficulty with GANs is that the discriminator is often significantly more powerful than the generator, which can lead to the issues such as mode collapse and lower gradient performance.

This study proposed a novel architecture to address these deficiencies. They include a powerful spatial attention mechanism to control the generator’s operation. The custom-designed discriminator, in particular, analyses the probability of the image being real and fake an attention map based on that evaluation. This attention map reveals critical regions for differentiating between real and fake images as viewed by the discriminator model. The summer-to-winter, orange-to-apple, and horse-to-zebra datasets are used to evaluate model performance. The Kernel Inception Distance (KID) used as an evaluation metric, and the findings were analyzed as KID Apple to Orange 4.30 ± 0.31 and KID Summer to Winter dataset 1.18 ± 0.16. They show that the suggested technique outperforms previous state-of-the-art image translation frameworks in both qualitative and quantitative terms (Lin et al., 2021).

This study introduced a method to address mode collapse in GANs. The generator maps multiple inputs to the same output. They mitigated mode collapse through a virtual mapping approach. This approach integrates two processes, i.e., merge and split, to enhance the diversity. Before training the discriminator, the merge step merges many data points into one. As a result, the generator learns to capture the distribution of merged data rather than the distribution of individual, unmerged data points. After the training, the generator’s output undergoes a splitting process to separate its contents and generate diverse modes. The MNIST, 2D-ring, and 2D-grid datasets are used in this study. VirtualGAN covers all modes and generates high-resolution samples (Abusitta, Wahab & Fung, 2021).

The proposed model differentiates between real and fake images and retouching GAN-generated images. Their experiments achieved better accuracy in recognizing GAN-generated images. The study presented an innovative strategy to deal with mode collapse, which employs i.e., a set of generators, an encoder, and a discriminator. The proposed formulation simultaneously trains all of these components. The orthogonal vector technique is used in conjunction with numerous generators learning separate information (Jain et al., 2020).

The study proposed the Multi-Generator Orthogonal GAN. They involve passing the created synthetic data through an encoder, which extracts feature vectors. The orthogonal value between these feature vectors are then calculated. The values accurately represent the degree of correlation between the vectors, indicating how diverse the information gathered by the generators. The smaller orthogonal values indicate that the generators have learned a wider variety of knowledge. These orthogonal values are combined with the initial generator loss to update the parameters of the individual generators collectively.

The extensive experiments performed on MNIST, CIFAR10, and CelebA datasets to evaluate the MGO-GAN’s performance, were demonstrating considerable improvements in data quality. The CIFAR10 dataset obtained IS 6.130 and FID 198.894, MNIST IS 7.454 and FID Score, and CelebA FID 189.188. The model successfully mitigates vanishing gradient problems during training process (Li et al., 2021a).

The proposed study advances a deep generative model to enhance synthetic media generation and detection capabilities. It employs data augmentation, label smoothing, and dropout layers in the discriminator to improve GAN performance. Experiments with different parameter adjustments identify optimal settings, resulting the study achieve FID score of 55.67 and 98.82% accuracy in detection.

Materials and Methods

The detailed flow of research process is illustrated in Fig. 1. They perform pre-processing on the dataset image augmentation is applied on the training dataset while, enhancing the discriminator’s capacity for better model generalization. The augmented training data improved the discriminator and generator model which helps to enhance its capacity to differentiate between real and fake images. There are various evaluation measures used in this research to analyze the model’s efficacy. These metrics are used to assess the quality of the generated images as well as the model-detecting capabilities.

Figure 1 Detailed flow of research process.

This image is sourced from the Flickr Faces Dataset (https://www.kaggle.com/datasets/dullaz/flickrfaces-dataset-nvidia-128x128; License: CC BY-NC-SA 4.0). Face 2 was randomly selected from the GAN model generated fake faces.

Dataset

The research work used the Flickr-Faces Dataset (Nvidia) which includes 70,000 human faces with image sized 128 × 128 and the Fakefaces dataset with 1,400 images generated by StyleGan2. Both datasets are available on Kaggle (see Data Availability section).

Image preprocessing

The proposed model applies various image transformation techniques to preprocess input data before it is fed into the model. Each image is first resized to 128 pixels and then centered and cropped to 64 pixels, with a focus on the central region. This phase is important for removing unnecessary background material and increasing the relevancy of the visual data. The images are converted into tensors. Tensors serve as the basic data structure for input and output inside the deep learning structure. Normalization is used to standardize pixel values, with the first tuple (0.5, 0.5) representing mean values across each channel RGB and the second tuple (0.5, 0.5) representing channel-wise normal deviations. This normalization procedure improves the deep learning model’s convergence during training. These transformation changes prepare the image data for optimal deep learning model training by providing a standardized size, transforming it into an appropriate data structure to improve overall model performance.

Dataset splitting

The dataset is divided into training and testing sets 68,600 images are used for model training and 2,800 images are used for model testing. The training set is used to train the model and the testing set is essential in evaluating the model’s performance and measuring its generalization capabilities.

Image augmentation

The research work uses image augmentation to artificially increase the size and diversity of the training dataset while enhancing and improving the model generalization ability. The brightness levels are adjusted between 0.9 and 1.1 which affects the total light intensity in the image increasing brightness for a brighter image appearance. Contrast transformation increases the differentiation between light and dark regions. Saturation, which influences color purity which is modified within a range of 0.9 to 1.1, impacting color richness. Hue, represents color forms such as red, green, and blue, moves over the full spectrum ranging from: −0.1 to 0.1 causing changes in the overall color tone of the dataset images.

Architecture

The detection and generation of synthetic media poses a critical challenges due to the increasing difficulty in distinguishing between real and fake images. This research work proposed a robust GAN model that excels at both generation and detection tasks. The study employs a combination of data augmentation techniques, label smoothing, adds a 10% dropout layer within the discriminator block, and careful model fine-tuning into the architecture of DCGAN as illustrated in Fig. 2. This approach aims to create a robust model in generating realistic images and detecting real and fake images. The GAN model is made up of two main parts i.e., a Generator and a Discriminator. The generator is fed a latent vector, also referred to as random noise, and outputs an image as a result. In this study, a latent vector of size 52 is employed for noise generation. The meaningful latent space learning is critical for the model to increase generalization and allow the model to efficiently reflect data changes. The Generator architecture employs transposed convolutional layers (ConvTranspose2d) for up sampling by enhancing the spatial resolution of the input data.

Figure 2 Architecture of DCGAN.

The Generator’s input feature map is set to 64. Padding is employed in transposed convolutions, and a padding value of one is used to regulate the size of the output feature map. To handle color images, the number of output channels is set to three, and the kernel size for the transposed convolution filter is set to four. Batch normalization is used before the activation function in the neural network architecture to improve training stability and speed. The rectified linear unit (ReLU) activation function is used, which replaces negative values with zero to introduce non-linearity while avoiding the vanishing gradient problem, and also retaining computing efficiency.

Tanh activation function is used in the Generator’s last layer to ensure that the output pixel values are between [−1, 1]. The real images often have pixel values in the [0, 255] range which allows for a simple linear mapping of produced pixel values to the real image range through scaling and shifting. The data augmentation technique are used to diversify the dataset and improve the Generator’s performance. This augmentation improves the dataset, allowing the Generator to cover more data modes.

The proposed GAN model, with these architectural modifications and training methodologies, intends to robustly address the issues of synthetic media detection. The discriminator works in the same way as the generator, evaluating 64 * 64 * 3 input images and returning a single probability value indicating whether the input is real or fake. The discriminator’s input feature map size is set at 64. The architecture is made up of a series of convolutional layers that use LeakyReLU activations to overcome the “dying ReLU” issue and ensure a low, non-zero gradient for negative inputs.

Utilizing a Dropout layer implies that around 10% of the input units will be stochastically zeroed during each training iteration. This regularization technique assistance in preventing overfitting by introducing stochasticity, thereby encouraging the network to learn more generalized features. This prevents neurons from becoming inactive during training. The proposed model parameters settings are presented in Table 1. The output of the D is flattened and passed through a Sigmoid activation function, generating a probability score. This score represents the probability that the input image is real or fake. The label smoothing technique implemented during training involves replacing the conventional binary labels of 1 and 0 for real and fake data, respectively, with softened values, where real data is set to 0.9 and fake data is set to 0.2. The G and D models have a weight initialization function. This function, designed for convolutional layers, generates weights by putting values from a normal distribution with a mean of 0 and a standard deviation of 0.02. The function sets weights for batch normalization layers to have a mean of 1.0 and a standard deviation of 0.02, for bias set to 0 (Radford, Metz & Chintala, 2016). This weight initialization strategy is consistently applied to both the generator and discriminator models. It ensures that the model parameters start from values that encourage stable and effective learning during the training process for both the G and D components of GANs.

Table 1 Model training specification.

Model specification	value	
Model execution	Nvidia Tesla P100 GPUs on Kaggle platform.	
Input resolution	128 × 128	
Learning rate (Generator)	0.0001	
Learning rate (Discriminator)	0.0001	
Optimizer	RMSprop	
Epochs	25	
Iteration per epoch	500	
Batch size	256	
Dropout	10%	
Latent-vector	52	

The backpropagation is used by both G and D networks to update the model depending on the computed loss. This repeated process of changing model weights allows both components to enhance their performance across training epochs, boosting the GAN overall capabilities.

Experimental settings

The numerical tests were conducted using the Kaggle platform, utilizing the acceleration provided by NVIDIA TESLA P100 GPUs to improve the model’s training speed. The implementation was carried out using Python 3.10.12, and the PyTorch Version 2.0.0 package were employed for the model development. The RMSprop optimizer is used to train the GAN, with different learning-rate for the G (0.0001) and D (0.0001). Discriminator is trained using a batch size of 256 samples. Discriminator use activation function LeakyReLU with a slope of 0.2. The model trained for a total of 25th epoch to achieves optimal results. The model generate synthetic images with dimensions 64 × 64 pixels and three color channels. The training process includes saving generated images at every 500 iterations at the end of each epoch.

Results

In order to assess the performance of GANs, one approach involves calculating the divergence between the marginal distribution of real images and the conditional distribution of fake images (Fathallah, Sakr & Eletriby, 2023). The feature distance of images is measured using a pre-trained Inception v3 network, a classifier built on the ImageNet dataset as a feature extractor. The Inception v3 model extracts features from both real and fake images to calculate FID, allowing for an extensive feature-by-feature comparison. The lower FID score indicating that the generated images closely reflect a distribution of real images. The real images known as ground truth. FID analysis both real and fake images, resulting in a comprehensive judgment by including realistic visuals. This complex method of FID, includes a comparison with real and fake images replicating better results.

Table 2 compares DCGAN, WGAN and Proposed model on Flickr Faces Nvidia dataset on the bases of Fréchet Inception Distance. The proposed GAN outperforms other models while achieving an FID score of 55.67.

Table 2 Fréchet Inception Distance (FID) scores of different models on Flickr Faces Nvidia dataset.

Model	FID score	
DCGAN	88.75	
WGAN	99.46	
Improved DCGAN
(Proposed model)	55.67	

Figure 3 shows the generate output of DCGAN, WGAN, and the proposed model at the 25th epoch presenting a graphical representation of the outcomes obtained by each model. In terms of synthetic media detection, the proposed model outperforms other models as illustrated in Table 2. When compared to other models, the evaluation metrics are obtained from the testing set, illustrating the robustness of the proposed model achieving 98.82% accuracy in synthetic media detection.

Figure 3 Synthetic images of (A) DCGAN; (B) WGAN; (C) proposed model.

This image is sourced from the GAN model Generated face file. These images have been randomly taken from the generated images.

The proposed model demonstrates better performance with a recall of 0.99, precision of 0.99, highlighted in Table 3. The graphic representation of the receiver operating characteristic (ROC) curve includes the true positive rate (TPR) which represents the proportion of actual real cases the model correctly identifies as shown in Eq. (1) and the false positive rate (FRP) represents the proportion of actual fake cases as shown in Eq. (2) (Murray & Rawat, 2022).

Table 3 Precision, recall, accuracy and F1-score of different models.

Model	Precision	Recall	Accuracy	F1-score	
DCGAN	0.81	0.72	77.82	0.77	
WGAN	0.66	0.60	64.54	0.63	
Improved DCGAN
(Proposed model)	0.99	0.99	98.82	0.99	

(1) TPR=TruePositives(TP)TruePositives(TP)+FalseNegatives(FN)

(2) FPR=FalsePositives(FP)TrueNegatives(TN)+FalsePositives(FP)

ROC curve of models, illustrates in Fig. 4 the TPR (sensitivity or recall) on the y-axis and the FPR on the x-axis proposed model achieve 0.99 ROC-AUC. TP analyses the proportion of real images correctly detected by the model are 1,384, whereas FPR assesses the fraction of fake images incorrectly categorized as real is 17. ROC curve effectively demonstrates the dynamic trade-off between these rates across different categorization thresholds, providing an extensive overview of the model’s discriminating performance.

Figure 4 ROC Curve of different models (A) proposed model; (B) WGAN; (C) DCGAN.

Figure 5 presents a comparison of FID (Fréchet Inception Distance) and accuracy metrics among DCGAN,WGAN and the proposed model. The findings show that the proposed model outperforms both DCGAN and WGAN in terms of FID and accuracy. This improved performance demonstrates its capacity to generate realistic images while also accurately distinguishing between real and fake ones.

Figure 5 Comparing FID score and accuracy of different models.

A confusion matrix is a model performance evaluation method providing a detailed analysis of a classification model’s accuracy. It displays counts for true positives, true negatives, false positives, and false negatives, presenting an entire overview of the model’s performance. The proposed model performs better, with greater percentages of true positives and true negatives shown in Fig. 6. This implies that the model successfully identifies real cases and accurately recognizes fake instances as well.

Figure 6 Proposed model confusion matrix.

Figure 7 presents different model evaluation measures which provide a comprehensive evaluation of each model’s performance. These measures indicate the models’ capacity to distinguish between real and fake images, providing valuable insights into their discriminatory abilities.

Figure 7 Evaluation metrics of different models.

The generator loss is significantly higher than the discriminator loss, which indicates that G struggles to fool D, maybe leading to convergence issues. If the discriminator loss is significantly smaller than the G loss, discriminator may consistently output 0 for all fake images, limiting the generator from creating low-quality images. An efficient GAN architecture attempts to keep balance D and G losses within a comparable range (Fathallah, Sakr & Eletriby, 2023). As illustrated in Fig. 8A shows that the training process of the DCGAN model was unstable compared to both the Wasserstein GAN and the proposed GAN, as depicted in Figs. 8B and 8C. Accuracy serves as a metric, quantifying the frequency at which a model accurately predicts outcomes. It is computed by dividing the count of correct predictions both true positives and true negatives by the total number of predictions calculated as shown in Eq. (3) (Murray & Rawat, 2022). Figure 9 illustrated the proposed model training accuracy at 25th epoch and testing accuracy.

Figure 8 Generator and discriminator loss during training (A) DCGAN losses; (B) Wasserstein GAN; (C) proposed model losses.

Figure 9 Proposed model training and testing accuracy.

(3) Accuracy=NumberofCorrectPredictionsTotalNumberofPredications

The study conducted experiments shown in Table 4 to fine-tune the proposed model for optimal parameter settings. The study intentionally chooses parameter combinations in which the model excels at balancing G and D loss and achieving better results. The adjustment of learning rate for the G and D changing batch sizes, Latent-Vector and optimizers to fine-tune the model. The objective is to find the best parameters for the model.

Table 4 Proposed model performance tuning with different batch size; learning-rate; latent-vector and optimizer.

Learning rate	Latent-vector	Batch size	Optimizer	FID	Accuracy	
G 0.0001
D 0.0001	52	256	Adam	211.81	99.46%	
G 0.0001
D 0.0001	52	256	Adadelta	391.55	98.71%	
G 0.0001
D 0.0001	52	256	RMSprop	55.67	98.82%	
G 0.0001
D 0.0001	52	256	SGD	365.42	97.89%	
G 0.0001
D 0.0001	52	128	Adam	120.62	99.54%	
G 0.0001
D 0.0001	52	128	Adadelta	387.17	98.14%	
G 0.0001
D 0.0001	52	128	RMSprop	62.92	41.39%	
G 0.0001
D 0.0001	52	128	SGD	361.51	95.71%	
G 0.0001
D 0.0001	100	256	RMSprop	56.71	85.93%	
G 0.0001
D 0.0001	200	256	RMSprop	55.32	69.43%	
G 0.0001
D 0.0001	64	256	RMSprop	429.43	100.00%	
G 0.0001
D 0.0001	100	256	Adam	254.17	99.89%	
G 0.0001
D 0.0001	200	256	Adam	277.45	99.93%	
G 0.0001
D 0.0001	64	256	Adam	255.41	98.93%	
G 0.0001
D 0.0001	100	256	Adadelta	363.99	99.39%	
G 0.0001
D 0.0001	200	256	Adadelta	361.37	99.14%	
G 0.0001
D 0.0001	64	256	Adadelta	365.21	99.32%	
G 0.0001
D 0.0001	100	256	SGD	351.28	98.93%	
G 0.0001
D 0.0001	200	256	SGD	361.19	98.86%	
G 0.0001
D 0.0001	64	256	SGD	354.61	98.86%	

Conclusions and future work

The study introduces advancements to a deep generative model aimed at significantly enhancing its capabilities in both synthetic media generation and detection. Employing a comprehensive approach, the research incorporates data augmentation techniques and label smoothing strategies to improve the performance of GANs. Additionally, the integration of dropout layers within the discriminator block further refines the model’s performance. The study conducts experiments involving various parameter adjustments including optimizer selection, batch size modification, adjustment of latent vector, and learning rate optimization the study seeks to identify optimal settings to maximize the model’s effectiveness. The results indicate that the proposed model excels in both image manipulation and detection tasks. They demonstrate better performance by achieving lower FID score 55.67 and high accuracy 98.82%. Moreover, comprehensive evaluations assess the efficacy across various measures to ensure the model effectiveness. Future work will concentrate on improving the model’s robustness in generating high-resolution images and training on datasets featuring multiple human faces in a single image to enhance real and fake detection capabilities. The study limitations include the current model’s challenges in handling variations in lighting conditions. Additionally, exploring the scalability of training on larger datasets will be investigated to further refine and generalize the model’s performance, which could impact its accuracy in real and fake detection.

Supplemental Information

Supplemental Information 1 Code.

Additional Information and Declarations

Competing Interests

Author Contributions

Data Availability

The authors declare that they have no competing interests.

Rabbia Zia conceived and designed the experiments, performed the experiments, performed the computation work, prepared figures and/or tables, authored or reviewed drafts of the article, and approved the final draft.

Mariam Rehman conceived and designed the experiments, performed the experiments, performed the computation work, prepared figures and/or tables, authored or reviewed drafts of the article, and approved the final draft.

Afzaal Hussain conceived and designed the experiments, analyzed the data, prepared figures and/or tables, authored or reviewed drafts of the article, and approved the final draft.

Shahbaz Nazeer performed the experiments, analyzed the data, authored or reviewed drafts of the article, and approved the final draft.

Maria Anjum analyzed the data, performed the computation work, authored or reviewed drafts of the article, and approved the final draft.

The following information was supplied regarding data availability:

The Fakeface and Flickr-Faces (Nvidia) datasets are available in Kaggle:

- https://www.kaggle.com/datasets/hyperclaw79/fakefaces

- https://www.kaggle.com/datasets/dullaz/flickrfaces-dataset-nvidia-128x128.

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
