# Peer review of "Improving synthetic media generation and detection using generative adversarial networks"

_PeerJ Computer Science, doi:10.7717/peerj-cs.2181_

## Round 0.1 · original submission · Major Revisions

Dear authors,

Thank you for submitting your article. Feedback from the reviewers is now available. It is not recommended that your article be published in its current format. However, we strongly recommend that you address the issues raised by the reviewers, especially those related to readability, experimental design and validity, and resubmit your paper after making the necessary changes.

Best wishes,

**Language Note:** PeerJ staff have identified that the English language needs to be improved. When you prepare your next revision, please either (i) have a colleague who is proficient in English and familiar with the subject matter review your manuscript, or (ii) contact a professional editing service to review your manuscript. PeerJ can provide language editing services - you can contact us at [email protected] for pricing (be sure to provide your manuscript number and title). – PeerJ Staff

Reviewer 1 ·

Basic reporting

The authors present an improved GAN model for human faces. The data processing pipeline consists of image preprocessing, dataset splitting, image augmentation and data processing. A publicly available dataset, specifically Flickr-Faces, was used for the experiments and validation of the improved model. The title of the article does not fit the content of the paper. Whereas the title is holistic and rather leads to the assumption that it portrays a systematic review, the article itself is very specific by dealing with synthetic data for human face features. We would recommend specifying the title.
The structure of the article is coherent. The introduction gives an understandable motivation for the need for action. However, the introduction is repetitive as the functionality and structure of a GAN is explained multiple times (ll.40, ll.45 and ll.51) and the state of the art is anticipated superficially (ll.58-ll.60).
The reviewers highly recommend the authors improve the language as well as formal aspects. The introduction of abbreviations is conducted multiple times in an unstructured way (e.g. ll. 40, ll.45, ll.13, ll.51, ll.67, ll83). Sentences to not finish sensibly (ll.56, ll.178), upper and lower case are incorrect (ll. 66). Sentences are grammatically incorrect. Lastly, the use of space between words, references and remarks in brackets is not consistent (e.g. ll.44, ll.88, ll.128, ll. 156).
References in the state of the art should be improved by current research. Works from 2023 and 2022 relating to GAN are missing. The reference style is completely wrong. Please refer to the guidelines for authors.
The graphs fit the text sections and underline the findings. Nevertheless, they need some improvements. Specifically coherent writing and naming, colours, and layout. Graphs should not possess the same name, e.g. Fig. 4 Receiver Operating Characteristic (ROC) Curve or Fig. 8 generator Loss and Disciminator Loss. The quality of images is also to be improved as some parts of the images are cut off. For the visualization of architecture in Fig. 2, we propose known techniques as opposed to a flow chart.

Experimental design

The paper and the experiment fit the aims and scope of the journal. Some open questions remain. The authors claim an improved model but do not specify what specifically was improved and how. The findings include a rather descriptive analysis of the experimental results. A more in-depth analysis is required. A replication of the findings is possible with the code provided by the authors for Jupyter notebook. What framework and packages are used for implementation besides PyTorch. Information on training is missing, such as the number of epochs, weights initialization, learning rate adaption, early training stopping, etc.

Validity of the findings

The findings require a more in-depth analysis. Such as implications of false positives and false negatives, training curve and generated images. The contribution can be enhanced by stating the theoretical contribution of the experimental findings. Lastly, the section conclusion and future work lacks a detailed description of future research, shortcomings and the contributions of the paper.

Reviewer 2 ·

Basic reporting

Creating synthetic images is extremely harmful in terms of misleading the public, but it is a useful technique when conducting research on medical issues such as disease detection. I see this as an extremely important study in terms of filling the research gap on this subject. However, I will make a few corrective suggestions so that the work matures more and can be understood more easily by the readership.

1-"The current techniques frequently use generative adversarial networks (GANs) and encoder-decoder architectures for manipulated image generation and detection [9-22][32]."
It is not right to allocate 9-22 sources for a quotation like this. There are many references like this in the article. Remove all unnecessary resources.

2-In the article, it is stated that a GAN model that can detect with 94% accuracy has been developed. However, it is not clear how the improved DCGAN model was fully developed. It is not clear which layers were removed or what kind of processes were applied. In addition, it should also be stated what exactly the steps implemented for development will contribute to.

3- Data augmentation techniques often allow the model to overfitting. How can we understand that the GAN model developed here does not memorize?

Experimental design

1-Why was RMSprop used as the optimizer? Why was Adam optimizer not preferred?
2-What changes were made in the DCGAN model to reduce the FID value?
3-What exactly does a low FID value mean?
4-Figure 3 shows synthetic images produced with various GAN models. The "c" images in this figure were created by the model obtained with improved DCGAN and classified with 94% accuracy.

But isn't the importance of synthetic data the fact that it is extremely similar to real data?
What do you think Figure 3 c. How similar are the images produced to human face images? What is the benefit of capturing a 94% accuracy rate from these images?

Validity of the findings

1-The biggest innovation of the study is stated in the article that label smoothing and data augmentation techniques were used to increase the performance of the GAN. However, here it is necessary to express more clearly and understandably how the label is smoothing. The necessary mathematical expression must be given. His contribution to the development of the algorithm should be fully explained.
2-It was observed that the limitations of the article were not emphasized. This issue needs to be examined carefully.
3-It appears that no information is given about future studies.
4-It was observed that the conclusion part of the article remained quite narrow in scope. It is recommended to improve this part.
5-Sources need to be organized thoroughly. As can be seen in the first source below, "et al." The statement should be removed. All resources must be edited.
I. Goodfellow et al., Generative adversarial networks, Commun. ACM, vol. 63, no. 11, pp. 139144, 2020, doi: 10.1145/3422622.

---

## Round 0.2 · Minor Revisions

Dear authors,

Thank you for submitting your revised article. One reviewers did not accept the invitation to re-review the revision. Although one reviewer accepted your paper, it will be better to address the following for the quality:

1. Write the full term first, followed by the acronym or initialism in brackets. For subsequent mentions, use the abbreviated form rather than the full term. See for example the usage of GAN in the Abstract section.
2. Keywords should be written according to single style. They should also be listed in alphabetical order.
3. Blank character should be correctly used. It is untidily used/not used.
4. Organization of the paper should be presented at the end of Introduction section.
5. Use correct referencing style in the text.
6. The paper lacks the running environment, including software and hardware. The analysis and configurations of experiments should be presented in detail for reproducibility. It is convenient for other researchers to redo your experiments and this makes your work easy acceptance. A table with parameter settings for experimental results and analysis should be included in order to clearly describe them.
7. The authors should clarify the pros and cons of the methods. What are the limitation(s) methodology(ies) adopted in this work? Please indicate practical advantages, and discuss research limitations.
8. The research gaps and contributions should be clearly summarized in the introduction section. Please evaluate how your study is different from others in the related work section.
9. Equations should be polished. Explanation of the equations should be checked.

Best wishes,

Reviewer 2 ·

Basic reporting

no comment

Experimental design

no comment

Validity of the findings

no comment

Additional comments

no comment

---

## Round 0.3 · accepted · Accept

Dear authors,

Thank you for clearly addressing all the reviewers' comments. I confirm that the quality of your paper is improved. The paper now seems to be ready for publication in light of this revision.

Best wishes,